# A GNSS-based method to define athlete manoeuvrability in field-based team sports

**Grant Malcolm Duthie**[1]\*, **Sam Robertson**[2], **Heidi Rose Thornton**[3]

**1** School of Exercise Science, Australian Catholic University, Strathfield, New South Wales, Australia, **2** Institute for Health and Sport (IHES), Victoria University, Melbourne, Australia, **3** Gold Coast Suns Football Club, Metricon Stadium, Carrara, Queensland, Australia

\* grant.duthie@acu.edu.au

**Data Availability Statement:** The data underlying the results presented in the study are available directly from Kaggle: https://www.kaggle.com/duthieg/afl-nrl-tortuosity-data.

## Abstract

This study presented a method of quantifying the manoeuvrability of two field-based team sport athletes and investigated its relationship with running velocity during competition. Across a season, 10 Hz Global navigation satellite system (GNSS) devices were worn during matches by 62 athletes (Australian Football League [AFL]; n = 36, 17 matches, National Rugby League [NRL]; n = 26, 21 matches). To quantify manoeuvrability, tortuosity was calculated from the X and Y coordinates from match GNSS files (converted from latitude and longitude). Tortuosity was calculated as 100 x natural logarithm of the chord distance (distance travelled between X and Y coordinates), divided by the straight-line distance. The maximal tortuosity was then quantified for each 0.5 m·s⁻¹ speed increment, ranging from 0 to the highest value for each game file. A quadratic model was fitted for each match file, controlling for the curvilinear relationship between tortuosity and velocity. A comparison of the quadratic coefficients between sports, and within sport between positions was investigated using linear mixed models. Resulting standard deviations (SDs) and mean differences were then assessed to establish standardized effect sizes (ES) and 90% confidence intervals (CI). A curvilinear relationship exists between maximal tortuosity and running speed, reflecting that as speed increases, athletes' ability to deviate from a linear path is compromised (i.e., run in a more linear path). Compared to AFL, NRL had a greater negative quadratic coefficient (*a*) (ES = 0.70; 0.47 to 0.93) for the 5 second analysis, meaning that as speed increased, NRL athletes' manoeuvrability reduced at a faster rate than when compared to AFL. There were no positional differences within each sport. GNSS derived information can be used to provide a measure of manoeuvrability tortuosity during NRL and AFL matches. The curvilinear relationship between tortuosity and speed demonstrated that as speed increased, manoeuvrability was compromised.

## Introduction

The running demands of professional team sports are commonly quantified using variables derived from electronic player tracking systems such as global navigation satellite systems (GNSS). These devices include a GNSS chip that provides the position of the unit in space,

**Funding:** The authors received no specific funding for this work.

**Competing interests:** The authors have declared that no competing interests exist.

permitting the calculation of many movement variables including distance, speed, acceleration and metabolic power [1–3]. Additionally, some devices also include a tri-axial accelerometer, magnetometer and a gyroscope, permitting measurement of alternative non-running related variables such as PlayerLoad™, collisions and tackling [4,5], and more recently, change of direction angles [6]. Using GNSS devices, practitioners can quantify the external load of competition as it provides understanding of the required individual and positional demands of the sport, thus being useful in the prescription of training volume and intensity. Despite the value of this information, many GNSS systems only quantify running in one dimension (i.e., total distance, speed, acceleration, high speed efforts), which has limitations. Presently, there is a lack of information regarding how running in team sports is performed within two-dimensional space, comprising a large proportion of movements in team sports.

Team sports such as Australian rules football and rugby league are typically intermittent in nature, involving high intensity, high speed movements such as sprinting, rapid accelerations/ decelerations and changes of direction (COD) [7–10]. Anecdotally, these movements are used to evade opponents in attempts to score, or alternatively capture opponents, both often occurring at pivotal moments of the game. These actions are reactionary to the opposition's movements or position on the field, frequently occurring whilst running at a high velocity, or alternatively rapidly accelerating or decelerating [11]. Similarly to prey avoiding predators in the animal environment [12], these movements can require athletes to adopt a non-linear path (two dimensional), whilst maintaining speed and without jeopardizing control, also known as manoeuvrability [12]. Manoeuvrability has been extensively researched in other contexts including animal behaviour [13], aerodynamics [14] and vehicle design [15], but its applicability to sport is yet to be investigated.

Compared to straight-line running, changing direction whilst running at speed involves high magnitudes of vertical, mediolateral, and anterior-posterior impulses [16,17], placing higher mechanical load on lower limbs [18]. As highlighted in previous research [19], the ability to change velocity and/or direction in response to a stimulus is termed agility [20]. Since the reactive component cannot be measured using GNSS devices [19], only the physical component is applicable in this context. Therefore, a measure of manoeuvrability is differentiated from agility both in its definition and practical utility, which will be explored. Despite their large application in team sports for load monitoring and prescription, GNSS devices have not been used to quantify these non-linear movements in research, despite their extensive potential to do so. Potentially, the ability to accurately quantify such movements may result in practitioners implementing these into training programs more frequently and specific to the demands of competition.

Recently, a novel method to accurately identify and measure predetermined COD angles was established [6], using a multistage algorithm that incorporates triaxial inertial sensor inputs. Inertial sensor inputs included roll (mediolateral), pitch (anterior-posterior) and yaw (superior-inferior), providing information to detect precise COD movements for predetermined movements [6]. This algorithm provides scope to quantify rapid COD movements, however the signal processing methods required to utilise this approach would be difficult in applied settings as accelerometer data is more complex than GNSS X and Y coordinates. Alternatively, the manoeuvrability or diversity of animal movements has been quantified via the calculation of tortuosity [21]. In this context, manoeuvrability can be defined as a departure from a straight path, thus a random path being more tortuous than a straight line [21]. Practically, the same principles apply to team sports, where athletes manipulate their running depending on spatial constraints (i.e., needing to evade opposition), adopting a more tortuous route where necessary. Currently there is no research pertaining to the assessment of tortuosity in team sport athletes. It could be proposed that an athletes' ability to display tortuosity at higher

speeds may be a measure of their manoeuvrability, which would on first observation appear a highly advantageous physical capacity. Manoeuvrability can be considered an important physical capacity, as athletes frequently need to adopt a non-linear running path to evade or catch their opposition without jeopardizing their running speed or control. Potentially if manoeuvrability was appropriately targeted in training, it may be improvable and performance advantages may result.

In professional soccer, using GNSS devices, the relationship between athletes' maximum velocity attained and heading change (the difference in horizontal heading angle between two consecutive points in time) were measured, demonstrating trade-off between velocity and heading angles [19]. This research provides preliminary evidence of the use of GNSS devices to measure more complex movements athletes frequently undertake. However, investigating an athletes' manoeuvrability using tortuosity in a team sport environment may have important applications for individual program design and potentially even in assessing performance.

Therefore, the purpose of this study was to propose a measure of manoeuvrability known as tortuosity, and to investigate its relationship with running speed. A comparison between sports is presented, as well as some practical examples of the applications of tortuosity within team sports, such as demonstrating tortuosity between early and late-stage rehab compared to a match. It was hypothesized, based on previous studies regarding animal behaviors [21], that the tortuosity of team sport athletes would decrease as running speed increased.

## Materials and methods

### Design and subjects

Physical activity profiles were measured during the 2020 professional National Rugby League (NRL) and the 2020 professional Australian Football League (AFL) seasons using GNSS devices. Both the NRL and AFL seasons comprise of weekly games over a 24- to 26-week duration. From GNSS devices, raw 10 Hz data were exported using manufacturer provided software (detailed below) and longitude and latitude were converted to X and Y coordinates.

Twenty-six male rugby league (age = 25.4 ± 4.1 years; stature = 187.4 ± 6.4 cm; body mass = 100.4 ± 9.8 kg) and 36 male Australian football (age = 23.9 ± 3.7 years; stature = 187.2 ± 7.7 cm; body mass = 85.9 ± 7.7 kg) athletes took part in this study from two clubs playing in the NRL and AFL competitions, respectively. For both teams, athletes from all positional groups were included (despite no positional analyses conducted). The AFL squad included; midfielders (n = 11), mobile backs (n = 4), mobile forwards (n = 9), ruck (n = 1), tall back (n = 4) and tall forwards (n = 3). For NRL, the squad comprised of edge forwards (n = 4), fullback (n = 1), halves and hookers (n = 6), middle forwards (n = 9) and outside backs (n = 6). Athletes were included in the study if they played a game for their respective team, and completed the match (i.e., were uninjured). Prior to and during the competitive season, athletes from both teams participated in a full-time professional training program. This entailed up to four field-based training sessions per week, undergoing specific skill-based training, as well as speed and conditioning training. Additionally, up to four resistance-based sessions were completed, with a primary focus on strength and power development. All data were collected as part of the routine monitoring processes of the club with athletes volunteering to provide their data for research purposes and data were deidentified prior to analysis. Prior to commencement of the study, ethical approval was sought by the Australian Catholic University Ethics Committee (2018-290E).

### Global navigation satellite systems and data analysis

GNSS devices were used to measure the physical activity profiles of players during 21 NRL (12 losses, 9 wins) and 17 AFL matches (11 losses, 5 wins, 1 draw) across the 2020 season. For

analysis, there were a total of 372 individual match files for AFL (2 were removed due to injury) and 342 for NRL, corresponding to a total of 714 match files for the sports combined. Matches were played weekly during the competition, with match recovery periods ranging from 4 to 10 days in AFL and 5 to 9 days for NRL. For AFL, the mean number of match files per player was 12 ± 6 (range; 1 to 17) and for NRL was 7 ± 6 (1 to 18). The same microtechnology units were used for both sports (Vector, Catapult Sports, VIC, Australia) which comprise a 10 Hz GNSS chip. The same device was worn by each athlete for both sports across the season, as to minimize interunit variability [22] and was fitted in a secure pouch sewn within the playing jersey. For both sports, jerseys were tight fitting to minimize measurement noise but is also common practice for devices to be worn in jerseys during matches rather than manufacturer provided vests.

Devices were switched on prior to the warm-up (~20 minutes) allowing adequate time for satellite lock to be achieved. If an athlete was unable to complete the match due to injury, their respective match file was removed prior to analysis. Following the completion of matches, devices were downloaded using proprietary software (Openfield, Catapult Sports, VIC, Australia) and individual player match file was exported in raw form (10 Hz) into a comma delimited file (csv.). These files provide a range of information including speed, acceleration, latitude, longitude, satellite count and data quality (horizontal dilution of position [HDOP]). For AFL games, there was an average HDOP of 0.61 ± 0.05 and 13.12 ± 0.83 satellites. For NRL games, there was an average HDOP of 0.78 ± 0.15 and 13.79 ± 1.64 satellites.

For each match file, latitude and longitude were converted to X and Y coordinates using the *geospatial* package within the RStudio program (V 1.3.1056). The X and Y coordinates were filtered using a 4th order, 1 Hz low pass Butterworth filter. A 1 Hz cut-off filter was employed following a visual inspection of the residual analysis of cut off frequencies between 0.1 to 10 Hz, as previously described [23]. The "rolling" distance travelled by the athlete between X and Y coordinates over a one, two, five and 10 second duration was established and was termed 'chord distance'. The rolling method involves including the current X and Y coordinates, and then rolls through the length of the file. To investigate the effect of duration on tortuosity values, a one, two, five, and 10 second period was adopted, as it was expected that these time frames were long enough to allow players to reach a high speed but was also long enough to negate the effect of any short, low speed movements. The mean running speed over the examined duration was also established. A rolling 'straight distance' was established by calculating the straight line distance between the current X and Y coordinate and the X and Y coordinate at the beginning of the assessment period. Tortuosity was then calculated as 100 x natural logarithm of the chord distance divided by the straight line distance. A tortuosity value of 0 represents moving in a straight line (i.e., linear) while any value greater than 0 occurred when an athlete deviates from a linear path, expressed as a percentage. The maximal tortuosity was then quantified for each speed from 0 to the highest value for match file in 0.5 m·s⁻¹ increments. Fig 1 is an example of an athletes' tortuosity over a five second duration during a low speed (~2.5 m·s⁻¹) run involving a change of direction. Fig 2 is an example of an individual athletes' tortuosity values over the entire duration of a single game, with the maximal values at each 0.5 m·s⁻¹ interval displayed.

## Statistical analysis

Visual inspection of the natural logarithm of maximal tortuosity (i.e., log[100 x log(chord distance/linear distance)]) for each running speed revealed a quadratic relationship (curvelinear). Subsequently, a quadratic model was fitted to the data points for each individual player for each game, controlling for the curvilinear relationship between tortuosity and speed. The

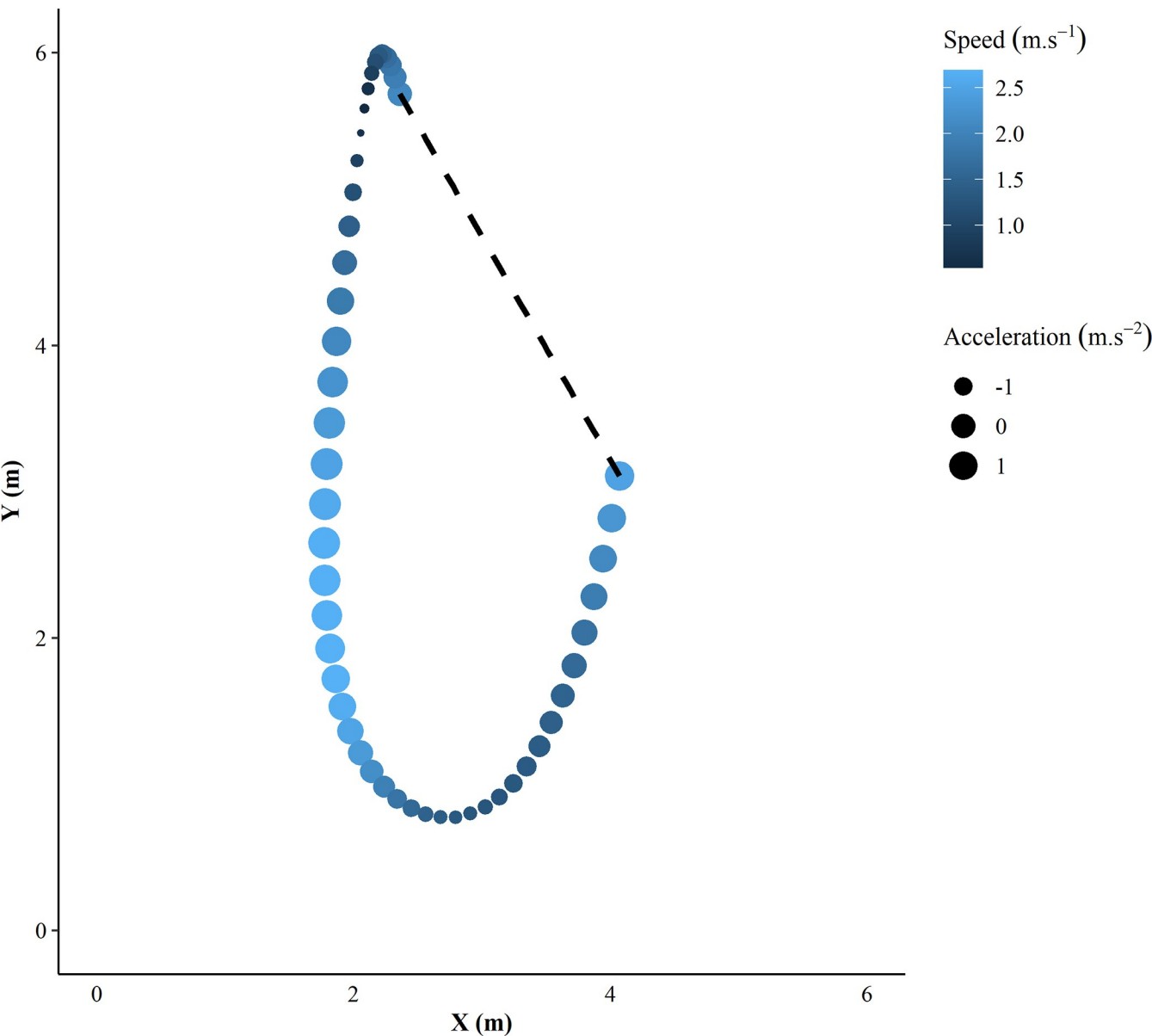

**Fig 1. An example of the tortuosity of a 5 second movement (50 data points) for a single athlete, with the speed and acceleration of the movement also shown.** In this example, the athlete covered 9 metres in total, with 3 metres between the X and Y coordinate. This resulted in a tortuosity value of 111%.

quadratic models included a fixed effect (the intercept; natural log of tortuosity), the predictor (running speed) and the square of the predictor, which together estimate the mean quadratic effect. The mean ± SD $R^2$ value for the models was 0.96 ± 0.04. For each athletes' individual game file, the quadratic coefficient ($a$), linear coefficient ($b$), and intercept ($c$) were established. Specifically, $a$ represents the overall position of the curve up and down the y axis (i.e., wide or narrow), $b$ reflects the upward or downward linear trend in y values along the x axis, and $c$ is a constant (intercept), representing where the relationship sits on the y axis. To examine the ability of the model to distinguish between sports, a linear mixed model using a random intercept design were used. In this model, athlete identification was included as a random effect, the fixed effect was the sport, and the predictor was either the quadratic coefficient ($a$), the

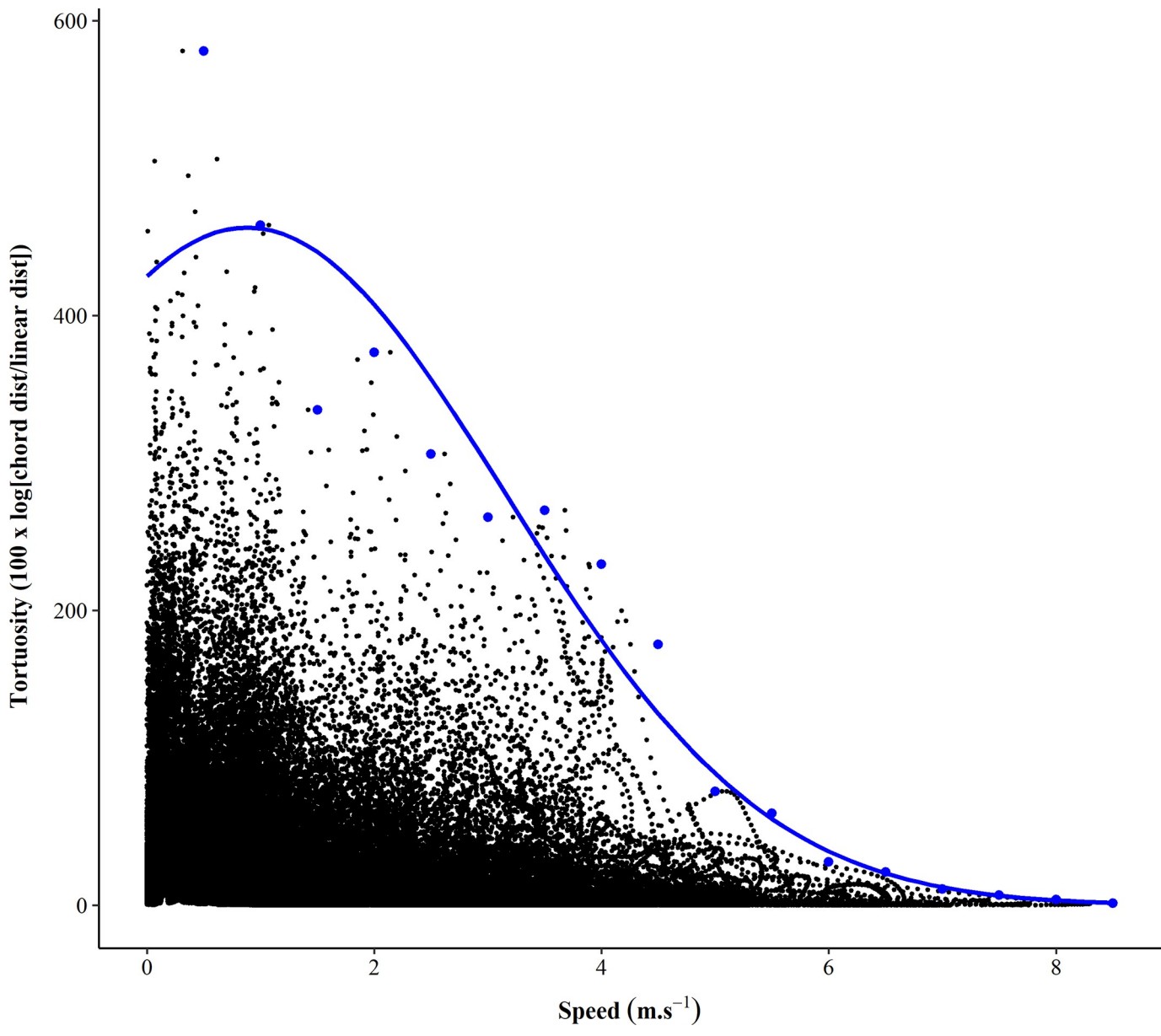

**Fig 2. An example of the raw tortuosity (10 Hz) for a single athletes' match file established over a 5 second duration.** The blue dots represent the maximal tortuosity value observed in each 0.5 m·s⁻¹ interval, while the blue line represents the quadratic model fitted to the data.

linear coefficient (*b*), or the intercept (*c*) obtained from the quadratic model. The least squares mean test was used to compare between sports and resulting SDs and mean differences were then assessed to establish standardized effect sizes (ES) and 90% confidence intervals (CI). Standardized effect sizes were described using the magnitudes; <0.20 trivial; 0.21–0.60 small; 0.61–1.20 moderate; 1.21–2.0 large and > 2.01 very large [24]. Effects were deemed to be real if they were 75% greater than the moderate worthwhile difference (calculated as 0.6 x the between-athlete SD) for reasons previously described [25,26]. All statistical analysis was performed in R Studio software (version 1.3.1093, RStudio Inc.)

**Table 1. Depicts the quadratic model summary, providing the mean ± SD quadratic coefficient (*a*), the linear coefficient (*b*) and the intercept (*c*) from the models for each duration.**

| Sport | Duration (s) | Quadratic coefficient (a) | Linear coefficient (b) | Intercept (c) |
|---|---|---|---|---|
| AFL (n = 124) | 1 | -0.09 ± 0.06 | -0.17 ± 0.46 | 5.99 ± 0.53 |
| | 2 | -0.09 ± 0.05 | -0.02 ± 0.35 | 6.04 ± 0.39 |
| | 5 | -0.14 ± 0.06 | 0.46 ± 0.38 | 5.75 ± 0.40 |
| | 10 | -0.17 ± 0.07 | 0.65 ± 0.38 | 5.66 ± 0.39 |
| NRL (n = 171) | 1 | -0.11 ± 0.09 | -0.25 ± 0.55 | 6.18 ± 0.58 |
| | 2 | -0.13 ± 0.08 | -0.02 ± 0.47 | 6.15 ± 0.50 |
| | 5 | -0.20 ± 0.09* | 0.59 ± 0.45 | 5.72 ± 0.45 |
| | 10 | -0.20 ± 0.12 | 0.67 ± 0.55 | 5.65 ± 0.47 |

*depicts a difference greater than 0.6 SD compared to AFL for the same duration.

## Results

Table 1 depicts the quadratic model summary, providing the mean ± SD quadratic coefficient (*a*), the linear coefficient (*b*) and the intercept (*c*) from the models for each duration. Between sports, there was a substantial difference in the quadratic coefficient (*a*) (ES = 0.70; 0.47 to 0.93) for the 5 second duration analysis, with no other differences were evident. Fig 3 demonstrates the curvilinear relationship between running speed and tortuosity by sport (AFL vs NRL) as well as the duration of the analysis (a one, two, five, and 10 second periods).

## Discussion

This study presented a novel method of measuring the manoeuvrability of two field-based team sport athletes. Manoeuvrability can be considered an important physical capacity, as athletes frequently need to adopt a non-linear running path to evade or catch their opposition without jeopardizing their running speed or control. The primary finding of this study was that the measurement of tortuosity presents as a practical method to assess the manoeuvrability of athletes that can be calculated using commonly collected GNSS data. Our hypothesis was partially supported whereby there was a decrease in tortuosity as running speed increased. This finding has implications for training prescription and rehabilitation and although not examined here, potentially performance evaluation in some sports. The relationship between manoeuvrability and running speed was investigated across multiple durations. These durations included one, two, five and ten seconds, where a curvilinear relationship was identified between the maximal tortuosity and increased increments in running speed. It was demonstrated that NRL and AFL athletes typically complete non-linear movements peaking at a speed of 2 m·s⁻¹, decreasing thereafter to approximately 5 m·s⁻¹ where tortuosity plateaus, and a linear running path is adopted (Fig 3). An advantage of the methods proposed in this paper are that tortuosity can be calculated simply using the X and Y coordinates derived from GNSS data.

Although this is the first study to investigate tortuosity in team sports, this concept has been investigated extensively in other fields, particularly regarding animal behaviours regarding catching prey [21]. Whilst there are alternate ways to quantify tortuosity (i.e., fractal dimensions, sinuosity index etc.) [27,28], the method presented in this study represents a analysis for those with appropriate expertise that could be implemented using GNSS data. In the context of AFL and NRL, tortuosity can be used to quantify a large range of movements, such as complex accelerations, decelerations and rapid changes of direction. These movements require athletes to deviate from running in a linear path (as shown in Fig 1). There is a known trade-off

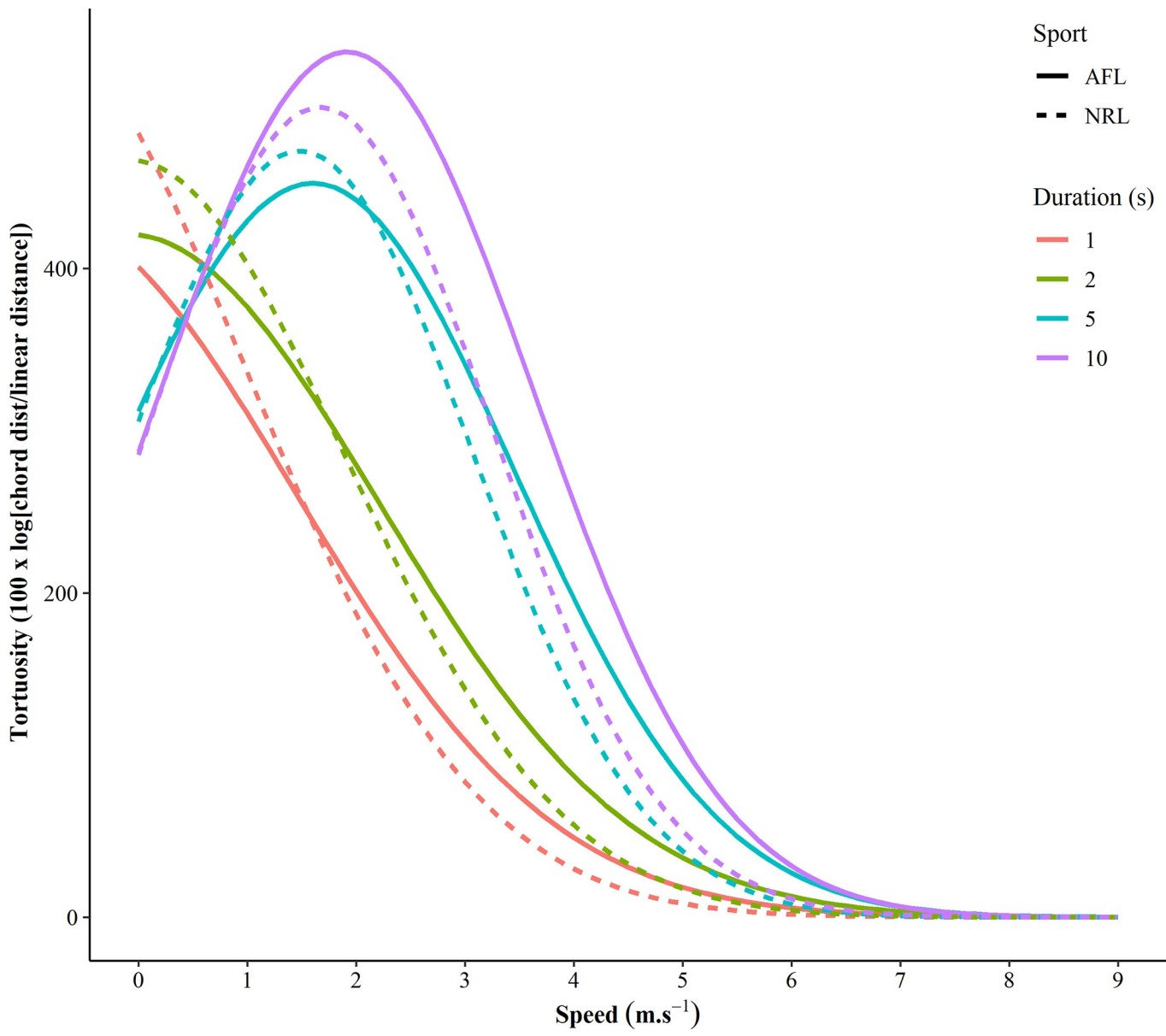

**Fig 3. A demonstration of the curvilinear relationship between running speed and tortuosity separated by sport and the duration of analysis.** Data represents the mean tortuosity.

between tortuosity and speed in other animal species [27], a relationship that can be referred to as manoeuvrability which is a favourable physical ability of athletes to possess. This novel study investigated the relationship between manoeuvrability and running speed, where Fig 1 demonstrates that there is a large proportion of low manoeuvrability events within competition across the speed spectrum. As seen in Figs 2 to 4, there was a curvilinear relationship between running speed and manoeuvrability for both AFL and NRL. This reflects that as running speed increases, manoeuvrability decreases, requiring athletes to adopt a more linear path at higher running speeds. This is a comparable finding to other research within professional soccer [19]. Achieving higher manoeuvrability at lower speeds is an expected finding, as within both sports athletes often have two feet in contact with the ground and are in close proximity

to an opponent or are trying to evade opponents when the ball is in dispute. At lower speeds, sharper COD movements (greater heading angle) are likely to occur, therefore a higher tortuosity will be evident.

Table 1 depicts the model coefficients for both AFL and NRL which together represent the relationship between running speed and tortuosity. Specifically, the *a* (quadratic) coefficient reflects the shape or position of the relationship. In this study NRL athletes demonstrated a substantially greater negative *a* coefficient, representing that as speed increased, NRL athletes' manoeuvrability reduced at a faster rate than when compared to AFL (also depicted in Fig 3). This finding may simply reflect differences in the objectives each game, where in NRL higher speeds are often associated with break in play (i.e., attackers breaking the defensive line). Once this occurs, athletes are generally attempting to reach the try line as quickly as possible, which where possible, a straight line run will be performed as defending players will be behind them [29]. In contrast, AFL is played on a large oval field resulting in a more free-flowing game, generally with players dispersed across the entire ground as teams attempt to implement zone or full-ground team defence [30]. Given this spread of athletes across the field, when athletes perform high velocity movements, anecdotally, there is a greater opportunity to run in curved paths to evade the opposition. This allows time for other attacking players to move into free space.

Despite the known differences in the purpose and aims of each sport, in the context of manoeuvrability, both sports demonstrated a sharp decline following the peak manoeuvrability at 2 m·s$^{-1}$. This speed is where humans generally transition from walking to running [31]. As this movement involves a flight time, this decreases the ability to change direction due to a period involving no contact with the ground. This finding reflects that although there is an unavoidable trade-off between manoeuvrability and running speed, the upper bounds of this measure were not examined here, and potentially if this physical attribute was appropriately trained, this may be improvable. In Table 1, the *b* coefficient represents the position of the relationship across the x axis, where the shorter duration analysis had a negative *b* coefficient and the higher duration have a positive coefficient. As also depicted in Fig 3, this means that for shorter durations (one and two sec), as speed increases, tortuosity decreases faster than that of the longer analyses (five and 10 sec). These findings are logical, as longer time frame allows a higher speed to be ran, as well as an increased opportunity to deviate from a linear path, therefore shifting the relationship further right on the x axis. To the authors best knowledge, no research has provided data demonstrating the duration of individual high-speed running efforts in either AFL or NRL. As such, this study selected these varied durations (one, two, five and ten seconds) in an attempt to encompass true high speed efforts (longer durations) as well as shorter efforts (shorter durations) reflecting rapid changes of direction.

As depicted in Fig 3, the tortuosity of the short duration efforts (one and two seconds) was lower than that of the longer durations (five and ten seconds) decreasing in an almost linear manner as speed increased. In Table 1, the *c* coefficient also represents this finding, as a higher *c* shows that tortuosity is higher at a lower velocity, which was evident for the shorter duration analyses, compared to the longer durations. As a longer duration is needed to accelerate to attain a higher speed, this finding was expected. For the longer durations (five and ten seconds), the relationship depicted a true curvilinear relationship, where tortuosity peaked at around 2.5 m·s$^{-1,}$ decreasing thereafter. Similarly, within Fig 3 evident differences can be seen between sports for the same duration. Notably, at shorter durations, the tortuosity within NRL at lower speeds was higher, likely reflecting that shorter, rapid accelerations are undertaken when compared to AFL. Conversely, within AFL at longer durations, tortuosity was higher along the speed spectrum, which was an expected finding given that within AFL there is a greater opportunity to run at higher speeds for longer given the free-flowing nature of the

sport. For future analysis, it could be suggested that an adaptive duration is employed, whereby tortuosity is quantified over short durations when acceleration is high, but over longer durations when speed is high. This method may account for the varying physical components of team sports.

Rapid and frequent changes in speed (i.e., acceleration and deceleration) are common in team sports [2], therefore, it is crucial athletes are appropriately prepared for this in terms of physical capacity. This study presented a novel method to quantify changes in direction relative to running speed in two different professional team sport populations. It was identified that quantifying deviations from linear running are important when quantifying the mechanical loading on the body. This information can be utilized in the prescription of training, ensuring that training adequately prepares athletes for competition, but is also important in the return to play process. Anecdotally, athletes are exposed to a gradual increase in running intensity during the return to play phase [32] where GNSS variables such as speed, acceleration and PlayerLoad™ are commonly used. However, within the research there is typically an emphasis placed on linear running metrics [33]. Conversely, quantifying the manoeuvrability of players during this phase would assist in exposing them to the loadings that may be experienced during competition. Subsequently, Fig 4 depicts the tortuosity at each speed using a five second duration to compare an early and late stage NRL rehab run compared to a game. This figure provides a clear example of the direct application of determining tortuosity particularly in the context of returning to play. In the early stage rehab, tortuosity peaks at the lowest speed, demonstrating that as speed increased, a straighter path is adopted. Conversely, the late stage rehab run demonstrates that a higher tortuosity was attained at a faster speed. Although the quantification of high speed running is important, stabilising forces enhance speed during linear movement, but turning or changing direction requires destabilising forces [13]. From an applied perspective, using the coefficients depicted in Table 1, practitioners can determine the maximal tortuosity displayed in competition for a given running speed, calculated as:

$$Tortuosity = (ax^2 + bx + c)^e$$

This maximal tortuosity value could be used by practitioners as a return to play key performance indicator, by returning to pre-injury manoeuvrability without compromising speed (see Fig 4). From an injury prevention perspective, training for maximal manoeuvrability could also be implemented into training programs, alike other physical capacities. Given that high speed efforts that deviate away from a linear path result in greater joint loading [18], these repeated high forces are likely to trigger mechanobiological tissue responses of the muscles, tendons, ligaments, bones and cartilage [17]. As per other physiological responses to exercise, athletes need to be prepared, perhaps so far as being exposed to a particular level of tortuosity at a range of running speeds. It may be that by using specific training modalities (i.e., agility training, small-sided games, etc) that higher manoeuvrability may be achieved for a given speed in comparison to what was observed during competition. If athletes were to improve this ability, then perhaps they could display this physical trait during competition. Given GNSS is regularly used in training, a similar tortuosity versus speed analysis could be implemented for each player for each individual drill. It may be suggested that incorporating subtle changes of direction to increase tortuosity would then have direct application to the running patterns that occur in a game.

While the validity of GNSS devices for quantifying the speed of team sport athletes has been established [34], a known limitation of the present study is the limited information regarding the validity of using X and Y positional data from GNSS devices. One study [35] identified that there was a mean difference between GPS determined geodetic point of 1.08 ± 0.34m. The

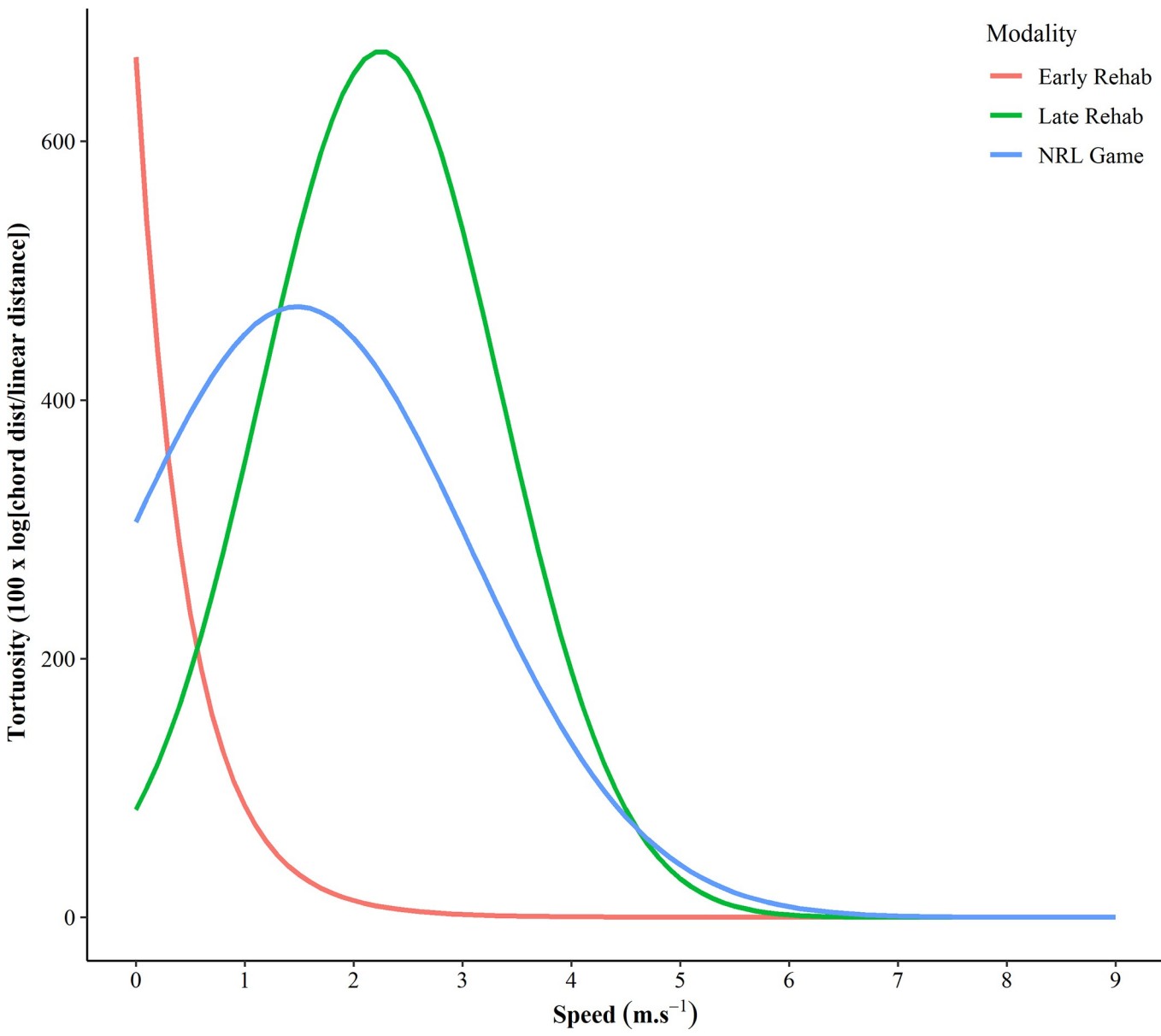

**Fig 4. A demonstration of the tortuosity at each speed using a five second duration to compare an early and late stage NRL rehab run compared to a game.**

small standard deviation evident here demonstrated that there was a biased error, however it was quite stable. This finding provides confidence in the methods used within the present study, although more research could be conducted using more recent GNSS devices, which record at higher frequencies compared to that used within other research [35].

In conclusion, the method presented in this study provides a novel assessment of the manoeuvrability of athletes from two team sports with data commonly collected from GNSS devices. The use of GNSS devices is widespread within both professional and semi-professional team sports, however, most variables provided by these devices are derived from speed-based metrics (i.e., high speed running distance, acceleration counts, metabolic power). When quantifying the demands of competition, assessing the training performed by athletes, or

monitoring the rehabilitation process; the evaluation of tortuosity may provide another aspect regarding the training and assessment of the athletes.

## Acknowledgments

The authors of this research wish to thank Professor Will Hopkins for his guidance in selecting an appropriate statistical analysis and also Professor David Martin for his assistance in developing this concept.

## Author Contributions

**Conceptualization:** Grant Malcolm Duthie, Sam Robertson, Heidi Rose Thornton.

**Data curation:** Grant Malcolm Duthie, Heidi Rose Thornton.

**Formal analysis:** Grant Malcolm Duthie, Heidi Rose Thornton.

**Funding acquisition:** Grant Malcolm Duthie.

**Investigation:** Grant Malcolm Duthie, Sam Robertson, Heidi Rose Thornton.

**Methodology:** Grant Malcolm Duthie, Sam Robertson, Heidi Rose Thornton.

**Project administration:** Grant Malcolm Duthie.

**Resources:** Grant Malcolm Duthie.

**Software:** Grant Malcolm Duthie, Heidi Rose Thornton.

**Supervision:** Grant Malcolm Duthie.

**Validation:** Grant Malcolm Duthie.

**Visualization:** Grant Malcolm Duthie, Heidi Rose Thornton.

**Writing – original draft:** Grant Malcolm Duthie, Heidi Rose Thornton.

**Writing – review & editing:** Grant Malcolm Duthie, Sam Robertson, Heidi Rose Thornton.

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
