## [Decision Letter · Decision Letter 0]

14 Jul 2021

PONE-D-21-13310

A method to define athlete manoeuvrability in team sports

PLOS ONE

Dear Dr. Duthie,

Thank you for submitting your manuscript to PLOS ONE. After careful consideration, we feel that it has merit but does not fully meet PLOS ONE’s publication criteria as it currently stands. Therefore, we invite you to submit a revised version of the manuscript that addresses the points raised during the review process.

We look forward to receiving your revised manuscript.

Kind regards,

Daniel Boullosa

Academic Editor

PLOS ONE

Journal Requirements:

Reviewers' comments:

Reviewer's Responses to Questions

**Comments to the Author**

1. Is the manuscript technically sound, and do the data support the conclusions?

Reviewer #1: Yes

Reviewer #2: Partly

2. Has the statistical analysis been performed appropriately and rigorously? 

Reviewer #1: Yes

Reviewer #2: I Don't Know

3. Have the authors made all data underlying the findings in their manuscript fully available?

Reviewer #1: Yes

Reviewer #2: Yes

4. Is the manuscript presented in an intelligible fashion and written in standard English?

Reviewer #1: Yes

Reviewer #2: Yes

5. Review Comments to the Author

Reviewer #1: The study is interesting and provides important and new information around the studied subject. It is very well designed, and the methods are quite clearly presented. However, it is important to provide more details about the variables/metrics analyzed since they are not usual in sport science studies. The outcomes become clearer in the discussion section, which was well constructed, but the results section should better explore and detail the data analyzed. A final paragraph with a brief summary of the conclusive statements and possible implications is required at the end of the study.

Abstract

The abstract was well developed and provides a good summary of the study. However, instead of explaining how each variable was calculated, it is important to describe “what” the variables represent. For example, the authors showed that “Compared to AFL, NRL had a greater negative quadratic coefficient…” but it is not clear what represents a “negative quadratic coefficient.” Is it related to good or bad manoeuvrability? This is essential to provide readers with a basic view of the rationale behind the study.

It is also important to briefly describe how many matches of each sport were analyzed.

Introduction

Lines 18-19? It is important to provide a reference to support this statement.

Lines 19-21: The “gamespeed” concept of Prof. Ian Jeffreys could be helpful to better support this idea (please, see some references below).

Jeffreys, I., Huggins, S., & Davies, N. (2018). Delivering a gamespeed-focused speed and agility development program in an English Premier League Soccer Academy. Strength & Conditioning Journal, 40(3), 23-32.

Jeffreys, I. (2010). Gamespeed: Movement training for superior sports performance. Coaches Choice.

Lines 22-24: This comparison is interesting, but maybe it would be better to use a reference more related to the sport science context. The “gamespeed” concept can also be used in this regard.

The introduction section was well designed, but before describing the purpose and objectives of the study, it is important to better emphasize the importance of assessing manoeuvrability in team sports and the practical relevance of this study to increase the body of knowledge on this topic.

Methods

It is important to better characterize the subjects/teams analyzed, including more information regarding them. Were they from the first division? How many games did they participate during the analyzed season?

Line 105: Please, delete “has been”.

Results

It is not completely clear what represents the “quadratic coefficient (a), the linear coefficient (b), or the intercept (c)” in relation to the manoeuvrability capacity. These analyses are not widespread and easy to understand (especially in sport science). Therefore, authors should better explain how these results are related to the study purposes. This is also essential to improve the impact and usefulness of the study for researchers and practitioners.

In the abstract section authors state that “A curvilinear relationship exists between maximal tortuosity and running speed, reflecting that as speed increases, athletes’ ability to deviate from a linear path is compromised.” However, this information is not clearly presented in the results section. It is important to provide more practical information for readers given the complexity of your analyses.

Figure 3: Does this figure show data from a representative athlete or mean values? Please, clarify.

Figure 4: It is not clear why authors presented this figure. It is not related to the study purposes. Please consider removing it.

Since the outcomes presented here are not usual in sport science studies, it is important to provide some information about the reliability of the variables presented.

Discussion

Line 182: It is not simple and practical and requires great experience in data analysis.

Lines 183-185: “This has implications for training prescription and rehabilitation and although not examined here, potentially performance evaluation in some sports.” Based on the results, it is not possible to affirm this. Please consider reformulating this statement.

Lines 188-190: “It was demonstrated that team sport athletes typically complete non-linear movements peaking at a speed of 2 m.s-1, decreasing thereafter to approximately 5 m.s-1 where tortuosity plateaus, and a linear running path is adopted.” This information should be presented in the results section. It is not clear how authors obtained this outcome.

Line 197: Again, this is not simple and practical. Please consider reformulating this.

Lines 206-208: This sentence is difficult to follow. Please, consider rephrasing this.

Lines 260-262: Please provide a reference for this statement.

Line 265: I understand the importance of the discussion around the rehab process, but it is beyond the study purposes and the match analysis performed. As suggested for figure 4, consider removing this part.

A conclusion is required to summarize the main outcomes found and reinforce the practical relevance of the study.

Reviewer #2: Review manuscript number: PONE-D-21-13310

Title: A method to define athlete maneuverability in team sports

Comments and Suggestions for Authors

General comments

-The article “A method to define athlete maneuverability in team sports” aimed to presented a method of quantifying the maneuverability of team sport athletes and investigated its relationship with running velocity during competition in team sports. I acknowledge the authors on their commitment to conduction such a difficulty study since working with athletes it is always hard. Also, I am pretty sure this topic is interesting and, indeed, more studies are required. However, I have major concerns regarding your study design and manuscript (listed below) which I think you should address thoroughly. I would be happy to review an updated version of your manuscript. following my comments and suggestions.

1) In general, with respect to the authors (effort and choices) the manuscript lacks methodological depth. Moreover, while reading your manuscript I feel a little lost (there is a lot of unhelpful information) ... Sometimes you were talking about differences in tracking systems…After you spoke about team sports physical demands… Change of direction/ reactive. What is your main objective???It is hard to follow…. Lot of information in the introduction

2) I really want to know the main purpose of using Global navigation satellite system (GNSS)? is your study the first to use this kind of material? If yes (please give information about validity and reliability...); If no (What is the novelty of your study….)

3) I noticed you used the term” team sports” in the title but you used only (football and rugby)?? No other disciplines?? … I think that the title is very general and do not reflect your methodology.

4) Physical and physiological demands of team sports vary from one discipline to another and it is very hard and not logical to give a general conclusion only by relying on two disciplines… Perhaps you could orient your study objective to compare only the requirements of Football and Rugby… it will be more relevant in my opinion.

5) You did not give additional information about the state of health of the participants during the whole sports season (injuries, illnesses ... which can stop sports practice) and of course it will influence your results.

6) You did not give additional information about the sports calendar of the participants during the whole sports season (congested game phase…Playoff... Covid19 lockdown) each phase has its own physical requirements and of course it will influence your results.

Specific comments

Introduction

1)The authors needed to provide a much more robust justification for this study based on the importance of (using such materiel); (why only) and finally what is new in your work (differences with previous studies investigating).

2)I suggest to add a clear hypothesis at the end based on what was previously reported (in the literature).

Methods

1) Did you proceed with a priori power analysis?

2) Please add BMI and BF%

3) Please add players training program/ History…

4)Please add the players position (it is important!!)

5)Please add inclusion criteria…

Results

-The results of the manuscript are well presented (Nothing to address)

Discussion

1)By following my suggestions (concerning adding a clear hypothesis) the authors may support or reject this hypothesis in the first part of the discussion…

2) The authors provide the most relevant information about the topic in this part (good work). However, I advise the authors to avoid using long sentences… I suggest to reduce the length of this topic (please try to be selective) …

Conclusion

1)The most important question here is what is the novelty that your study gives to the field??

2) Please add a practical applications part in which you may clearly explain how does the present study affect the field.

6. PLOS authors have the option to publish the peer review history of their article (what does this mean?). If published, this will include your full peer review and any attached files.

Reviewer #1: No

Reviewer #2: No

---

## [Author Response · Author response to Decision Letter 0]

23 Aug 2021

Manuscript ID: PONE-D-21-13310

Title: A method to define athlete manoeuvrability in team sports

Comments to the editor

The authors wish to thank the editorial team for the thorough review of this manuscript. All the comments provided were extremely helpful, and as such we believe the amendments made have increased the quality of this manuscript.

Reviewer 1

Comments to the Author

Reviewer #1: The study is interesting and provides important and new information around the studied subject. It is very well designed, and the methods are quite clearly presented. However, it is important to provide more details about the variables/metrics analyzed since they are not usual in sport science studies. The outcomes become clearer in the discussion section, which was well constructed, but the results section should better explore and detail the data analyzed. A final paragraph with a brief summary of the conclusive statements and possible implications is required at the end of the study.

Response: The authors wish to thank you for your time in reviewing this manuscript, it is greatly appreciated. All comments were helpful, and the suggested amendments have improved the quality of this manuscript. We have addressed the main points you have stated. Thank you again. 

Abstract

The abstract was well developed and provides a good summary of the study. However, instead of explaining how each variable was calculated, it is important to describe “what” the variables represent. For example, the authors showed that “Compared to AFL, NRL had a greater negative quadratic coefficient…” but it is not clear what represents a “negative quadratic coefficient.” Is it related to good or bad manoeuvrability? This is essential to provide readers with a basic view of the rationale behind the study.

Response: Thank you for highlighting this with us. As we have tried to summarise a novel concept, we have had to include more detail regarding the methods as possible. However, we agree that insufficient information was included about what our findings mean. Some information about the calculation has been removed, and additional text has been included about what tortuosity represents, as well as in the results what a negative quadratic coefficient means. We hope these changes meet your expectations. 

It is also important to briefly describe how many matches of each sport were analyzed.

Response: Thank you for highlighting this to us. This information has now been included; “62 athletes (Australian Football League [AFL]; n = 36, 17 matches, National Rugby League [NRL]; n= 26, 21 matches).”

Introduction

Lines 18-19? It is important to provide a reference to support this statement.

Response: The authors have added the word ‘anecdotally’ here, as what we are discussing is more of a conceptual background of the movements that we are attempting to quantify. Further, we can’t find any research which specifically prove this statement, however our statement is logical in this context.

Response:

Lines 19-21: The “gamespeed” concept of Prof. Ian Jeffreys could be helpful to better support this idea (please, see some references below).

Jeffreys, I., Huggins, S., & Davies, N. (2018). Delivering a gamespeed-focused speed and agility development program in an English Premier League Soccer Academy. Strength & Conditioning Journal, 40(3), 23-32.

Jeffreys, I. (2010). Gamespeed: Movement training for superior sports performance. Coaches Choice.

Response: Thank you for this suggestion. Based on the contents of each article, we have included the first one as a reference for our statement regarding movements in the game as we thought this was most relevant here. 

Lines 22-24: This comparison is interesting, but maybe it would be better to use a reference more related to the sport science context. The “gamespeed” concept can also be used in this regard.

Response: We appreciate this suggestion, however as we are linking to the calculation of tortuosity and its application in other contexts (outside of sport) we believe it is most appropriate as It currently stands. As tortuosity has been researched in other context, we have emphasized this more, therefore have included an additional sentence; “Manoeuvrability has been extensively researched in other contexts including animal behaviour (13), aerodynamics (14) and vehicle design (15) but is also applicable in a sporting context.” We hope this will suffice.

The introduction section was well designed, but before describing the purpose and objectives of the study, it is important to better emphasize the importance of assessing manoeuvrability in team sports and the practical relevance of this study to increase the body of knowledge on this topic.

Response: Whilst we agree with your statement here, throughout the introduction we have stated the practical relevance of manoeuvrability in team sports in small parts, therefore it may not have stood out. However, based on the suggestions of yourself and reviewer 2, we have attempted to increase this by including additional sentences throughout. We hope this has improved the introduction. Two of these sentences read;

“Perhaps, the ability to accurately quantify such movements may result in practitioners implementing these into training programs more frequently and specific to the demands of competition.”

“Potentially if this physical attribute was appropriately trained, this may be improvable and performance advantages may result.”

Methods

It is important to better characterize the subjects/teams analyzed, including more information regarding them. Were they from the first division? How many games did they participate during the analyzed season?

Response: Whilst we thank you for this suggestion. Based on your suggestion as well as reviewer 2, we have added considerable detail here. Specifically, the competitions (AFL/NRL) are the names of the respective competitions (i.e., the division) of the respective sports, therefore the terminology of the division isn’t applicable here. Regarding the number of games, this information is detailed in the ‘Global Navigation Satellite Systems and Data Analysis’ section, as we deem this the most appropriate spot. We have also provided additional information relating to the athletes’ positions and their training programs. We hope this clears up your concern. 

Line 105: Please, delete “has been”.

Response: Thank you, this is deleted now.

Results

It is not completely clear what represents the “quadratic coefficient (a), the linear coefficient (b), or the intercept (c)” in relation to the manoeuvrability capacity. These analyses are not widespread and easy to understand (especially in sport science). Therefore, authors should better explain how these results are related to the study purposes. This is also essential to improve the impact and usefulness of the study for researchers and practitioners.

Response: We thank you for your comment here and we do understand your point. However, as results sections are simply for stating results and the discussion is intended to explain such results, we don’t believe here is appropriate to add any explanation. To help explain the findings, we have included a short explanation of what the quadratic coefficients mean, which has assisted in some explanation in the discussion section. This reads as “Specifically, a represents the overall position of the curve up and down the y axis (i.e., wide or narrow), b reflects the upward or downward linear trend in y values along the x axis, and c is a constant (intercept), representing where the relationship sits on the y axis.” We have discussed the applied nature of these findings in greater detail in the Discussion, where typical convention would dictate.

In the abstract section authors state that “A curvilinear relationship exists between maximal tortuosity and running speed, reflecting that as speed increases, athletes’ ability to deviate from a linear path is compromised.” However, this information is not clearly presented in the results section. It is important to provide more practical information for readers given the complexity of your analyses.

Response: Thank you for your comment here. We have included this information and further explanation of the findings in the discussion rather than the results. As per the journal guidelines, we have selected to have ‘separate’ results/discussion/conclusions, therefore any results explanation is situated within the discussion. As we understand your overall point, we have made substantial changes to the discussion to better link the findings to a practical interpretation. We hope these changes suffice. 

Figure 3: Does this figure show data from a representative athlete or mean values? Please, clarify.

Response: Thank you for highlighting this to us. This data is the mean by sport and duration - this is now included in the figure caption.

Figure 4: It is not clear why authors presented this figure. It is not related to the study purposes. Please consider removing it.

Response: Although this figure is not depicting an actual result as such of our study, this figure helps present the practical application of the use of tortuosity. As we are presenting a new concept (manoeuvrability) in this study, we have attempted to provide as many practical applications as possible, therefore we feel that including an example of a rehab session versus a game is important. We hope you understand our reasoning here. 

Since the outcomes presented here are not usual in sport science studies, it is important to provide some information about the reliability of the variables presented.

Response: Thank you for your comment here. The authors don’t fully understand what you’re referring to when you say ‘outcomes’. However, we assume you may be referring to the “validity” (i.e., accuracy) of GPS for monitoring players speed and position. As such, we have mentioned the validity of GPS for establishing XY position (line 311) and have also included a reference for the validity of GPS for monitoring speed. Please let us know if this is not what was being referred to.

Discussion

Line 182: It is not simple and practical and requires great experience in data analysis.

Response: The authors partially disagree with this statement, however, we have removed the word ‘simple’ from this sentence. The process involved is relatively straight forward for those involved or have experience using GNSS in applied settings. We agree that a degree of data analysis expertise is required, however this is a given within this applied cohort of practitioners.

Lines 183-185: “This has implications for training prescription and rehabilitation and although not examined here, potentially performance evaluation in some sports.” Based on the results, it is not possible to affirm this. Please consider reformulating this statement.

Response: Thank you for this comment. Whilst we take this comment on board, respectively we disagree, in that we are simply suggesting how tortuosity can be used beyond this paper in applied settings. As applied practitioners ourselves, this concept is one which we have already begun to implement within our programs, and tortuosity has provided us with an objective measure to quantify such movements. 

Lines 188-190: “It was demonstrated that team sport athletes typically complete non-linear movements peaking at a speed of 2 m�s-1, decreasing thereafter to approximately 5 m�s-1 where tortuosity plateaus, and a linear running path is adopted.” This information should be presented in the results section. It is not clear how authors obtained this outcome.

Response: Whilst we understand your point here, we have provided adequate information surrounding the curve-linear relationship between speed and tortuosity and have presented our findings of this in multiple ways (Table 1, Figure 2, 3 and 4). As we have presented a new concept here, we tried not to fixate over the statistical findings (i.e., differences between sports), rather by presenting and explaining the data in a practical way. As such, we have taken this comment on board and have added additional text within the discussion on the interpretation of the relationship coefficients in table 1. We believe these additions have strengthened this section, so we thank you again for this contribution. 

Line 197: Again, this is not simple and practical. Please consider reformulating this.

Response: We understand your perspective here, however as stated previously those working with GNSS data in professional settings require a standard of data analysis skills. To reflect this, we have included the wording “for those with appropriate expertise” into this sentence. We hope this addition is adequate. 

Lines 206-208: This sentence is difficult to follow. Please, consider rephrasing this.

Response: Thank you for highlighting this issue with us. We agree it was worded poorly and has since been reworded to be clearer. Thank you.

Lines 260-262: Please provide a reference for this statement.

Response: Thank you, in support of the statement we have added the following reference:

Hickey, J. T., Timmins, R. G., Maniar, N., Williams, M. D., & Opar, D. A. (2017). Criteria for progressing rehabilitation and determining return-to-play clearance following hamstring strain injury: a systematic review. Sports medicine, 47(7), 1375-1387.

Line 265: I understand the importance of the discussion around the rehab process, but it is beyond the study purposes and the match analysis performed. As suggested for figure 4, consider removing this part.

Response: Thank you for the comment. While we agree quantifying the rehabilitation process was not a purpose of the study, we feel that the example of using tortuosity in relation to speed provides a practical context of quantifying manoeuvrability in team sports when developing a return to play program. We wanted to highlight the ability of GNSS devices to quantify more than just speed, high speed running, and acceleration. As such, the authors have added some of this to the conclusion requested in the next comment.

A conclusion is required to summarize the main outcomes found and reinforce the practical relevance of the study.

Response: Yes, we agree that there should be a concluding statement here. As such, between lines 338 and 345 is a concluding paragraph. Thank you for this suggestion.

Reviewer #2: Review manuscript number: PONE-D-21-13310

Title: A method to define athlete maneuverability in team sports

Comments and Suggestions for Authors

General comments

The article “A method to define athlete maneuverability in team sports” aimed to presented a method of quantifying the maneuverability of team sport athletes and investigated its relationship with running velocity during competition in team sports. I acknowledge the authors on their commitment to conduction such a difficulty study since working with athletes it is always hard. Also, I am pretty sure this topic is interesting and, indeed, more studies are required. However, I have major concerns regarding your study design and manuscript (listed below) which I think you should address thoroughly. I would be happy to review an updated version of your manuscript. following my comments and suggestions.

Response: The authors wish to thank you for your time in reviewing this manuscript, it is greatly appreciated. All comments were helpful, and the suggested amendments have improved the quality of this manuscript. We have addressed the main points you have stated and hope that this meets your requirements. Thank you again. 

1) In general, with respect to the authors (effort and choices) the manuscript lacks methodological depth. Moreover, while reading your manuscript I feel a little lost (there is a lot of unhelpful information) ... Sometimes you were talking about differences in tracking systems…After you spoke about team sports physical demands… Change of direction/ reactive. What is your main objective???It is hard to follow…. Lot of information in the introduction

Response: Thank you for this comment. Based on this suggestion and that of reviewer 1, we have added some more content to the introduction to add clarity, in particular brief summarising sentences at the end of each paragraph. Each paragraph has a specific focus, and a purpose in relation to the new concept we are introducing in this manuscript. Regarding the differences in tracking systems, the introduction focuses solely on GNSS devices and no other tracking systems such as smart cameras and Local Position Systems, so we can’t provide any explanation for this point. Although there is a lot of information in the introduction, we feel that this is necessary as we are introducing a novel concept to team sports. In summary, our introduction includes a summary on; GNSS devices, movements within team sports, measuring these movements with GNSS, 

2) I really want to know the main purpose of using Global navigation satellite system (GNSS)? is your study the first to use this kind of material? If yes (please give information about validity and reliability...); If no (What is the novelty of your study….)

Response: GNSS devices are used extensively within team sports, both in a professional and research setting. As discussed in the manuscript, the validity of GNSS devices has been previously quantified. The novelty of the study is the quantification of manoeuvrability, which has not been done before in a team sport setting

3) I noticed you used the term” team sports” in the title but you used only (football and rugby)?? No other disciplines?? … I think that the title is very general and do not reflect your methodology.

Response: Thank you, yes we agree that team sports is quite general and as such have changed the title to “A method to define athlete manoeuvrability in field based team sports”? Further to this, we have changed parts of the discussion which refer to team sports and have changed this to ‘two field based team sports’. 

4) Physical and physiological demands of team sports vary from one discipline to another and it is very hard and not logical to give a general conclusion only by relying on two disciplines… Perhaps you could orient your study objective to compare only the requirements of Football and Rugby… it will be more relevant in my opinion.

Response: Thank you, and yes we agree that the demands of team sport vary, in particular between field and court based sports. Subsequently, and aligning with the title modification you suggested in your comment above, we have provided clarity throughout the paper in specifying that we were assessing two field based sports. We believe this has increased the quality of the manuscript. 

5) You did not give additional information about the state of health of the participants during the whole sports season (injuries, illnesses ... which can stop sports practice) and of course it will influence your results.

Response: We have clarified that the players were monitored over the standard competition phase which consists of weekly games. As we only monitored competition games, players that were deemed fit to play were only included in the analysis. We hope this clarifies your concern.

6) You did not give additional information about the sports calendar of the participants during the whole sports season (congested game phase…Playoff... Covid19 lockdown) each phase has its own physical requirements and of course it will influence your results.

Response: While we agree that changes to the competition schedule may modify the physical requirements of players, during this study we were lucky enough in Australia to continue with regular weekly competition games in the AFL and NRL. Subsequently, we have highlighted that games were on a weekly basis for the collection of data for this research project. Additional text relating to the match recovery periods (turn around length) has also been added, reading; ”Matches were played weekly during the competition, with match recovery periods ranging from 4 to 10 days in AFL and 5 to 9 days for NRL.”

Specific comments

Introduction

1)The authors needed to provide a much more robust justification for this study based on the importance of (using such materiel); (why only) and finally what is new in your work (differences with previous studies investigating).

Response: Thank you for this comment. Throughout the introduction, the authors believe there is sufficient detail relating to the importance of quantifying manoeuvrability. We have attempted to highlight the gap in research as much as possible. Specifically, we believe lines 25-27, 37-39, 51, 57-58 and lines 63-66 now include such information you have suggested. 

2) I suggest to add a clear hypothesis at the end based on what was previously reported (in the literature).

Response: Thank you for this suggestion. This has been included now. 

Methods

1) Did you proceed with a priori power analysis?

Response: In this context, a priori power analysis is not relevant as we are limited by the number of athletes within each squad, a factor in which we can’t control. Further, as we aren’t using inferential statistics to explain differences between the quadratic equation coefficients, this priori analysis again is not applicable. As our total dataset is comprised of 714 match files (372 for AFL and 342 for NRL), we are confident that our sample size is adequate. We hope you understand our reasoning here. 

2) Please add BMI and BF%

Response: In the context of both AFL and NRL, both BMI and BF% aren’t relevant anthropometric measures, and therefore not included in this study. Given the high muscle mass of these athletes compared to the general population, their BMI is typically categorised as ‘obese’ which is simply incorrect, and a limitation of this measure regarding extreme spectrums of muscle mass. Further, BF% is also non-applicable here as this cannot be measured reliability across the two sports, using the same tool (i.e., DEXA). Further, given the constraints applied to these athletes during the 2020 season and the restricted living and training environment, athletes weren’t able to have DXA scans or have skinfolds taken as this required additional staff or environments which weren’t permitted. As such, body mass and height are the most reliable metrics we have available which are included in this study. 

3) Please add players training program/ History…

Response: Thank you for this suggestion. We have included a section regarding their training programs, reading; “Prior to and during the competitive season, athletes from both teams participated in a full-time professional training program. This entailed up to 4 field-based training sessions per week, undergoing specific skills-based training, as well as speed and conditioning training. Additionally, up to 4 resistance based sessions were completed, with a primary focus on strength and power development.” Nevertheless, it is important to note that we are showing this method as an exemplar, and the actual position differences and sport differences are somewhat secondary (specifically because we haven’t looked at the performance side here or tried to predict tortuosity from a feature set (i.e., using BMI, training history, etc). We hope this explanation addresses the concerns of the reviewer.

4)Please add the players position (it is important!!)

Response: Thank you raising this oversight with us. Positions have bene included, reading; “For both teams, athletes from all positional groups were included (despite no positional analyses conducted). The AFL squad included; midfielders (n=11), mobile backs (n=4), mobile forwards (n=9), ruck (n=1), tall back (n=4) and tall forwards (n=3). For NRL, the squad comprised of edge forwards (n=4), fullback (n=1), halves and hookers (n=6), middle forwards (n=9) and outside backs (n=6).”

5)Please add inclusion criteria…

Response: As we have conducted this within a professional setting, there was no set inclusion criteria’ as this is simply if they completed game. However, we have included some text to clarify this, which reads “Athletes were included in the study if they played a game for their respective team, and completed the match (i.e., were uninjured).”

Results

-The results of the manuscript are well presented (Nothing to address)

Response: Thank you kindly for the positive feedback on this section of the manuscript

Discussion

1)By following my suggestions (concerning adding a clear hypothesis) the authors may support or reject this hypothesis in the first part of the discussion…

Response: Thank you. We have since included the statement; “Our hypothesis was partially supported whereby there was a decrease in tortuosity as running speed increased.” 

2) The authors provide the most relevant information about the topic in this part (good work). However, I advise the authors to avoid using long sentences… I suggest to reduce the length of this topic (please try to be selective) …

Response: Thank you for this suggestion. We have made this amendment throughout the discussion. 

Conclusion

1) The most important question here is what is the novelty that your study gives to the field??

Response: The authors believe we have added sufficient information relating to the novelty of this study, not only in the discussion but throughout the entire manuscript. Specifically, in the introduction, lines 25-27, 37-39, 210-212, 221 to 223, 296-297,303 -306, and throughout the concluding paragraph. Overall, we believe these sections (some of which are new to this revision) provide this information. Thank you again.

2) Please add a practical applications part in which you may clearly explain how does the present study affect the field.”

Response: Thank you for this comment. Reviewer 1 had the same suggestion, and as such between lines 320 and 327 is a concluding paragraph.

---

## [Decision Letter · Decision Letter 1]

8 Oct 2021

PONE-D-21-13310R1A method to define athlete manoeuvrability in field-based team sportsPLOS ONE

Dear Dr. Duthie,

Thank you for submitting your manuscript to PLOS ONE. After careful consideration, we feel that it has merit but does not fully meet PLOS ONE’s publication criteria as it currently stands. Therefore, we invite you to submit a revised version of the manuscript that addresses the points raised during the review process. Please, address the last reviewers' suggestions ASAP before acceptance.

We look forward to receiving your revised manuscript.

Kind regards,

Daniel Boullosa

Academic Editor

PLOS ONE

Journal Requirements:

Reviewers' comments:

Reviewer's Responses to Questions

**Comments to the Author**

1. If the authors have adequately addressed your comments raised in a previous round of review and you feel that this manuscript is now acceptable for publication, you may indicate that here to bypass the “Comments to the Author” section, enter your conflict of interest statement in the “Confidential to Editor” section, and submit your "Accept" recommendation.

Reviewer #1: (No Response)

Reviewer #2: All comments have been addressed

2. Is the manuscript technically sound, and do the data support the conclusions?

Reviewer #1: Yes

Reviewer #2: Partly

3. Has the statistical analysis been performed appropriately and rigorously? 

Reviewer #1: Yes

Reviewer #2: Yes

4. Have the authors made all data underlying the findings in their manuscript fully available?

Reviewer #1: Yes

Reviewer #2: Yes

5. Is the manuscript presented in an intelligible fashion and written in standard English?

Reviewer #1: Yes

Reviewer #2: Yes

6. Review Comments to the Author

Reviewer #1: Overall, the authors performed a good work after the first round of revision. There are still minor (but important) issues that must be addressed before the final acceptance of the study.

Lines 57-58: This sentence is not clear. What would be improvable? The running speed or the control? Please restructure the sentence for the sake of clarity.

The authors stated that they followed the journal guidelines to organize/divide the manuscript into results, discussion, and conclusions. Under this rationale, following the journal guidelines, the results section “should describe the results of the experiments”. Since figure 4 is not a “result” of the experiment, it should be removed from the study.

Accordingly, as figure 4 is not part of the results of the study, the discussion regarding this respective figure should also be deleted.

Reviewer #2: Review manuscript number: PONE-D-21-13310

Title: A method to define athlete maneuverability in team sports

Comments and Suggestions for Authors

General comments

first of all, I would like to congratulate the authors for their quality of response indeed the manuscript has been well checked according to my comments and suggestions ... However, I have three suggestions that I consider interesting and can give more originality and clarity in your work.

**Title

1) The title is very general in my opinion and do not show the originality of your work... I suggest to use this "Using GNSS as a valid tool to quantify maneuverability in australian football and rugby leagues"

** Objective

1)I suggest directing your objective to compare football and rugby in terms of maneuverability recorded by (GNSS) and the relation with physical requirements of each sporting discipline (maybe reported with time motions analysis)...

**Discussion

1) Suggest to discuss differences between football and rugby ( maneuverability GNSS) and compare your results with other using different method...

7. PLOS authors have the option to publish the peer review history of their article (what does this mean?). If published, this will include your full peer review and any attached files.

Reviewer #1: No

Reviewer #2: No

---

## [Author Response · Author response to Decision Letter 1]

20 Oct 2021

Response to Reviewers Comments

Lines 57-58: This sentence is not clear. What would be improvable? The running speed or the control? Please restructure the sentence for the sake of clarity.

Response: We have now restructured this sentence to read ‘Potentially if manoeuvrability was appropriately targeted in training, it may be improvable and performance advantages may result’

The authors stated that they followed the journal guidelines to organize/divide the manuscript into results, discussion, and conclusions. Under this rationale, following the journal guidelines, the results section “should describe the results of the experiments”. Since figure 4 is not a “result” of the experiment, it should be removed from the study.

Response: We understand your point as Figure 4 is not a direct ‘result’ from a specific hypothesis/experiment, however it is a crucial element in demonstrating the practical application of the findings. As PLOS One is an applied journal that is highly relevant to practitioners working with athletes, we have decided to present figure 4 in the discussion (line 301) to provide an exemplar of the method and provide direction for future research. 

Accordingly, as figure 4 is not part of the results of the study, the discussion regarding this respective figure should also be deleted.

Response: As mentioned above, we have moved figure 4 to the discussion and still believe that the associated discussion is relevant for the applied nature of PlosOne.

Reviewer #2: Review manuscript number: PONE-D-21-13310

The title is very general in my opinion and do not show the originality of your work... I suggest to use this "Using GNSS as a valid tool to quantify maneuverability in australian football and rugby leagues"

Response: We have considered the title and added the term ‘GNSS-based’ as we believe that this distinction further assists with the description of the paper. We do however see the potential for the method to be used in any running-based, outdoor team sports so would prefer that we did not delimit to the two exemplar sports on this occasion, so as to not limit its appeal, potential application and ability for citation elsewhere.

I suggest directing your objective to compare football and rugby in terms of maneuverability recorded by (GNSS) and the relation with physical requirements of each sporting discipline (maybe reported with time motions analysis)...

Response: We respectfully disagree as we believe that this is a secondary objective to the broader, more generalisable current aim. We use these two sports here to highlight the applicability of the method in two team sports, however the primary emphasis is on promotion of the method itself. Considering there is not a commonly reported (or from our practical experience, utilised) method similar to tortuosity we would maintain that it is justified that scope be broadly maintained as in the manuscript’s current format. 

Suggest to discuss differences between football and rugby (maneuverability GNSS) and compare your results with other using different method...

Response: We believe that we have already directly compared the two sports on lines 245-259 and again at lines 286-293. We are not entirely clear on what the reviewer is requesting in the second part of this comment, however if they would be willing to clarify then we will be happy to address.

---

## [Decision Letter · Decision Letter 2]

9 Nov 2021

A GNSS-based method to define athlete manoeuvrability in field-based team sports

PONE-D-21-13310R2

Dear Dr. Duthie,

We’re pleased to inform you that your manuscript has been judged scientifically suitable for publication and will be formally accepted for publication once it meets all outstanding technical requirements.

Kind regards,

Daniel Boullosa

Academic Editor

PLOS ONE

Additional Editor Comments (optional):

Reviewers' comments:

Reviewer's Responses to Questions

**Comments to the Author**

1. If the authors have adequately addressed your comments raised in a previous round of review and you feel that this manuscript is now acceptable for publication, you may indicate that here to bypass the “Comments to the Author” section, enter your conflict of interest statement in the “Confidential to Editor” section, and submit your "Accept" recommendation.

Reviewer #1: All comments have been addressed

Reviewer #2: All comments have been addressed

2. Is the manuscript technically sound, and do the data support the conclusions?

Reviewer #1: Yes

Reviewer #2: Partly

3. Has the statistical analysis been performed appropriately and rigorously? 

Reviewer #1: Yes

Reviewer #2: Yes

4. Have the authors made all data underlying the findings in their manuscript fully available?

Reviewer #1: Yes

Reviewer #2: Yes

5. Is the manuscript presented in an intelligible fashion and written in standard English?

Reviewer #1: Yes

Reviewer #2: Yes

6. Review Comments to the Author

Reviewer #1: (No Response)

Reviewer #2: Thank you for addressing my raised questions. Overall from my point of view, the paper is now clearer . Therefore, my recommendation to the editor is to accept this paper for publication.

7. PLOS authors have the option to publish the peer review history of their article (what does this mean?). If published, this will include your full peer review and any attached files.

Reviewer #1: No

Reviewer #2: No

---

## [Editor Report · Acceptance letter]

11 Nov 2021

PONE-D-21-13310R2 

A GNSS-based method to define athlete manoeuvrability in field-based team sports 

Dear Dr. Duthie:

I'm pleased to inform you that your manuscript has been deemed suitable for publication in PLOS ONE. Congratulations! Your manuscript is now with our production department. 

Kind regards, 

on behalf of

Dr. Daniel Boullosa 

Academic Editor

PLOS ONE